# Unveiling the adoption of metaverse technology in Bangkok metropolitan areas: A UTAUT2 perspective with social media marketing and consumer engagement

Chaloempon Sritong[1], Weerachon Sawangproh[2], Teerapong Teangsompong[1]*

1 Business Administration Program, School of Interdisciplinary Studies, Mahidol University (Kanchanaburi Campus), Kanchanaburi, Kanchanaburi Province, Thailand, 2 Conservation Biology Program, School of Interdisciplinary Studies, Mahidol University (Kanchanaburi Campus), Kanchanaburi, Kanchanaburi Province, Thailand

☯ These authors contributed equally to this work.

* teerapong.tea@mahidol.edu

**Data Availability Statement:** All relevant data are within the manuscript and its Supporting Information files.

## Abstract

As the metaverse emerges as a transformative digital realm, its adoption and integration into various aspects of society are subjects of increasing scholarly and practical interest. This research investigated the factors influencing the intention to use metaverse technology (IU) in Bangkok metropolitan areas, with a particular focus on the extended Unified Theory of Acceptance and Use of Technology 2 (UTAUT2) framework, alongside the role of social media marketing (SMM) and consumer engagement (CE). To verify behavioral intention, gender, age, and experience are proposed as moderating factors affecting the constructs on individuals' behavioral intention of metaverse technology usage. The study collected data from 403 Thai internet users living in Bangkok and its surrounding areas using an online questionnaire. Subsequently, the PLS-SEM method was employed to validate the research model's robustness and reliability. Structural model analysis revealed significant relationships among constructs, highlighting SMM's direct influence on UTAUT2 (β = 0.787) and CE (β = 0.211). Serial mediation analyzes demonstrated a fully mediating role of SMM influencing UI through CE (β = 0.572) and UTAUT2 (β = 0.306). Moderation analyzes revealed the association between SMM and IU, mediated through UTAUT2 and CE, is moderated by age and experience. Additionally, the integration of PLS-SEM and artificial neural network (ANN) models underscored the accuracy and predictive power of the proposed framework. The findings of this study not only contribute to academic literature but also offer practical implications for marketers aiming to navigate the metaverse landscape effectively. They emphasize the pivotal role of UTAUT2 constructs and the subtle interplay between SMM, CE, and IU in shaping successful marketing strategies.

**Funding:** The author(s) received no specific funding for this work.

**Competing interests:** The authors have declared that no competing interests exist.

## Introduction

Metaverse technology, a concept that has swiftly transcended science fiction into tangible digital reality, represents a paradigm shift in how individuals engage with the virtual world. Rooted in augmented and virtual reality, the metaverse converges physical and digital realms, creating immersive, interconnected experiences [1]. With the proliferation of advanced technologies, the metaverse has gained prominence across diverse sectors, from entertainment and gaming to education and business [2, 3]. The significance of metaverse technology lies in its potential to redefine the way we connect, collaborate, and consume information [4]. As digital spaces become more integral to daily life, the metaverse stands out as a transformative force with far-reaching implications for social interaction, commerce, and cultural experiences [5].

The significance of metaverse technology in the realm of social media marketing cannot be overstated, as it represents a groundbreaking evolution in how brands connect with their audiences [6]. The metaverse provides an immersive and dynamic platform for social interaction, allowing marketers to create compelling and personalized experiences [7, 8]. Through augmented and/or virtual reality elements, brands can engage users in ways previously unimaginable, fostering deeper connections and brand loyalty [9]. The metaverse's ability to blend the physical and digital worlds opens up innovative avenues for storytelling and product experiences, transcending the limitations of traditional social media formats [10]. As users increasingly seek more authentic and interactive online engagements, metaverse technology becomes a strategic frontier for marketers, enabling them to craft memorable and shareable content [6, 11]. Understanding and harnessing the potential of the metaverse is crucial for staying ahead in the ever-evolving landscape of social media marketing [6, 12], as it offers a transformative space where creativity, technology, and consumer engagement converge.

The existing literature on metaverse technology and social media marketing lacks insights into the intentions of Thai users. While some studies focus on consumer responses to virtual worlds and technology perception [13–15], there's a noticeable gap in understanding Thai users' acceptance of metaverse technology for marketing. Moreover, little is known about Thai user behavior, preferences, and challenges related to metaverse adoption. This gap hinders the development of tailored strategies for the Thai demographic and limits academic knowledge and business insights. Bridging this gap is crucial for informing effective marketing strategies in Thailand's dynamic digital landscape. Integrating user acceptance models can provide a comprehensive understanding of user motivations and challenges in adopting metaverse technology for social media marketing. Therefore, this research aims to investigate the intention of Thai users to adopt metaverse technology specifically within the realm of social media marketing. As metaverse technology becomes increasingly integrated into digital platforms, understanding user adoption behavior is crucial for marketers seeking to navigate this dynamic landscape [16]. The study seeks to uncover the factors influencing the adoption intentions of Thai users, providing valuable insights that can inform targeted strategies for social media marketing campaigns within metaverse environments. To guide the investigation, this research employs the extended Unified Theory of Acceptance and Use of Technology (UTAUT2) framework, a comprehensive model designed to understand user behavior in adopting new technologies [17]. UTAUT2 extends the original UTAUT framework by incorporating additional constructs such as hedonic motivation, price value, and habit [17]. By leveraging the extended UTAUT2 framework, this research aims to offer a nuanced understanding of Thai users' adoption intentions in metaverse environments. The insights garnered from this study have the potential to drive targeted and effective strategies for social media marketing campaigns tailored to the unique preferences and behaviors of Thai consumers in the digital landscape.

## Literature review

### Metaverse technology

Stephenson [18] introduced the term "Metaverse" as a conceptualization of a computer-generated universe. The metaverse refers to a virtual space seamlessly merging the physical and digital realms. In this environment, users possess avatars, analogous to their physical selves, allowing them to engage in an alternate life within virtuality—a symbolic representation of their real-world experiences [19]. In July 2021, Mark Zuckerberg, the CEO of Facebook, declared a significant shift in the company's strategic direction, moving away from its traditional role as a social media entity to position itself as a metaverse-focused company. Subsequently, in a second announcement later that year, it was revealed that Facebook would undergo a rebranding, adopting the new name Meta Platforms to reflect its commitment to and emphasis on the metaverse [20]. The evolution of metaverse technology has brought about a transformative shift in the digital landscape, creating immersive virtual environments that transcend traditional online interactions [6, 13]. As users increasingly engage with metaverse platforms, exploring its potential in various sectors becomes imperative [3]. Within this dynamic context, the literature surrounding metaverse technology has grown substantially [21], focusing on its applications, impact, and user experiences. Researchers have explored the potential of metaverse environments for social interaction, commerce, and entertainment, shedding light on the ways in which this emerging technology redefines digital engagement. In business, integrating metaverse technology into social media marketing enables participants to experience the immersive presence of the market, thereby creating interactive and lifelike experiences for their customers [6].

### Social media marketing (SMM)

Social media marketing, as a crucial component of contemporary digital strategy, has also been significantly impacted by metaverse technology [22]. The immersive and interactive nature of the metaverse presents new opportunities for brands to connect with their audiences. Studies within this domain delve into innovative marketing strategies within metaverse environments, examining how brands can leverage these platforms to create compelling and personalized experiences [7, 8]. The integration of metaverse technology into social media marketing practices represents a novel approach to consumer engagement and understanding user acceptance within this context becomes pivotal for businesses seeking to capitalize on this transformative trend [23].

### Consumer engagement (CE)

The notion of consumer engagement is a recent development in the field of marketing [24]. Consumer engagement refers to the dynamic and interactive relationship between consumers and brands, transcending traditional notions of transactional exchange. It encompasses the active involvement, interaction, and participation of consumers with a brand or its offerings [25]. This multifaceted concept involves creating meaningful experiences that resonate with individuals on emotional and practical levels. The importance of consumer engagement lies in its ability to build and sustain strong relationships between brands and their audience. Engaged consumers are more likely to be loyal, advocate for the brand, and actively participate in its community [26]. In the digital age, where communication with a vast audience is instantaneous and continuous, effective consumer engagement becomes a strategic imperative for businesses [27]. It not only enhances customer satisfaction but also contributes to brand loyalty, and positive word-of-mouth, and a competitive edge in the market [28]. As consumer

preferences and behaviors evolve, fostering meaningful engagement remains a key driver for brand success in today's dynamic business landscape.

## The Unified Theory of Acceptance and Use of Technology (UTAUT)

UTAUT has emerged as a foundational framework for comprehending individuals' decisions to accept and adopt new technologies. Initially proposed by Venkatesh et al. [29], UTAUT posits that four primary constructs—performance expectancy, effort expectancy, social influence, and facilitating conditions—directly influence behavioral intention and, consequently, behavior. These four constructs are subject to moderation by factors such as gender, age, experience, and the voluntariness of use.

**Performance expectancy (PE).** Performance expectancy is characterized by the extent to which a technology offers benefits to consumers in executing specific activities [29]. Perceived usefulness, extrinsic motivation, and job fit are three factors influencing performance expectancy, as outlined by Shin [30]. Within a competitive context, performance expectancy indicates whether an individual anticipates performing better or worse than their opponent [31]. Furthermore, highlighted that expectancy theories suggest that individuals are more likely to exert greater effort in a task if they have high expectations of improved performance as a result of their efforts [32]. Arpaci et al. [33] demonstrated a positive impact of performance expectancy on behavioral intention toward the metaverse.

**Effort expectancy (EE).** Effort expectancy relates to the ease with which consumers can utilize a technology [29]. This construct delves into the perceived simplicity of operation and user-friendliness required for the adoption and utilization of the technology. High levels of effort expectancy indicate that users believe the technology is easy to use and does not demand excessive mental or physical effort [34]. Conversely, if users find technology challenging or with a steep learning curve, it may lead to resistance and lower intentions to adopt the technology [35]. Therefore, effort expectancy plays a pivotal role in shaping users' behavioral intentions and, consequently, their actual usage behavior. It underscores the importance of making technology user-friendly and reducing perceived complexities to enhance the likelihood of user acceptance and adoption.

**Social influence (SI).** Social influence measures the degree to which consumers believe that significant individuals in their lives, such as family and friends, endorse the use of a specific technology [29]. Previous studies have verified that social influence contributes to the comprehension of behavioral intention in adopting an information technology e.g., Venkatesh et al. [29], Sim et al. [36], Lu et al. [37] provided additional details on social influence by defining it as an individual's aspiration to shape others' thoughts regarding the execution of a specific activity. Therefore, the influence of social networks and interpersonal relationships is crucial in understanding and predicting technology adoption behaviors within the framework of UTAUT.

**Facilitating conditions (FC).** Facilitating conditions encompass consumers' perceptions of the resources and support accessible to facilitate a particular behavior [29]. Users are more likely to anticipate adopting systems when they perceive the presence of resources and technical support [34]. A study by Wu et al. [38], which surveyed 394 respondents to assess 3G mobile communications services, demonstrated that facilitating conditions have a significant impact on behavioral intention.

## The Extended Unified Theory of Acceptance and Use of Technology (UTAUT2)

UTAUT2 is a comprehensive theoretical framework developed to enhance our understanding of the factors influencing individuals' acceptance and adoption of technology. Building upon

the original Unified Theory of Acceptance and Use of Technology (UTAUT), UTAUT2 integrates additional constructs (i.e., hedonic motivation, price value, and habit) to capture the complexities of user behavior in diverse technological contexts [39]. Hedonic motivation introduces the dimension of pleasure and enjoyment derived from using a technology, acknowledging the emotional aspects of adoption. Price value incorporates users' perceptions of the value associated with the financial costs or effort required for technology adoption. Habit emphasizes the development of routines or automatic behaviors linked to the use of a particular technology. In the UTAUT2 model, gender, age, experience [39] are proposed as factors that may influence or moderate the impact of these constructs on individuals' behavioral intention and technology use.

**Hedonic motivation (HM).**   Hedonic motivation refers to the enjoyment or pleasure derived from utilizing a technology, and studies, such as those conducted by Brown & Venkatesh [40], have demonstrated its significant role in influencing technology acceptance and usage. This dimension acknowledges the importance of the emotional and experiential aspects associated with technology adoption [41]. Hedonic motivation aims to offer participants a sense of self-fulfillment, allowing them to derive enjoyment and satisfaction when using the new system [42].

**Habit (HB).**   Habit is characterized as the degree to which individuals tend to carry out behaviors automatically due to prior learning [43]. Habits are influenced by the degree of interaction and familiarity with a technology, evolving at varied levels over time [39]. When a behavior is repeated, a habit is formed, and this habit strength is likely to influence the behavior, having a direct impact on technology use [44].

**Price value (PV).**   Price value is characterized as the cognitive tradeoff made by consumers between the perceived benefits of a technology and its associated monetary costs of usage [39]. In marketing research, the evaluation of price or cost is typically conceptualized in conjunction with quality to ascertain the perceived value of products or services. This approach reflects the interplay between individually perceived price-performance and personal financial considerations [44]. A favorable price value arises when potential consumers believe that the benefits of using a technology outweigh the associated monetary costs, thereby positively influencing the adoption and usage of the new technology by consumers [45]. In a study conducted by Aldossari & Sidorova [46], it was found that price value significantly influences behavioral intention in the context of smart homes.

## Behavioral intention (BI)

Behavioral intention denotes the extent to which an individual consciously plans their future actions [42]. It is a key concept in various theoretical models, including UTAUT, which seek to understand and predict human actions [47]. Individuals base their intentions on personal perceptions of usefulness and ease of use, and these intentions can serve as predictors for the future acceptance level of the technology [48]. The formation of behavioral intention is influenced by various psychological factors, including perceived value, performance expectancy, habit, social influence, effort expectancy, hedonic motivation, and more [49]. Therefore, behavioral intention plays a pivotal role in technology acceptance models as it is considered a precursor to actual behavior [50], providing insights into the likelihood of individuals adopting and using a particular technology in the future.

## Gaps in the literature related to Thai users' intentions to use metaverse technology for social media marketing

In the burgeoning landscape of metaverse technology, particularly concerning social media marketing, the existing literature highlights a notable research gap in understanding the

intentions of Thai users toward adopting this innovation for marketing purposes. While prior studies delve into consumer responses to branded virtual worlds, marketing implications, and the convergence of technology perception and online behavior related to metaverse use, there is a discernible void regarding insights tailored to the acceptance of metaverse technology for social media marketing among the Thai population [13–15]. Furthermore, the intricacies of Thai user behavior, preferences, and attitudes toward metaverse technology for social media marketing remain largely unexplored, along with limited attention given to potential challenges such as digital literacy, accessibility, privacy concerns, and security issues [51]. This gap in the literature hampers the development of effective strategies and interventions tailored to the unique needs of the Thai demographic. Bridging this gap is crucial not only for advancing academic knowledge but also for informing businesses and marketers navigating the dynamic digital terrain in Thailand. Addressing these gaps can lead to more culturally informed and effective strategies in leveraging metaverse technology for social media marketing within the Thai market. As metaverse technology gains traction, integrating user acceptance models into the discourse becomes essential for deciphering the motivations and intentions of users in adopting this innovative technology, providing a comprehensive foundation for understanding challenges, opportunities, and user dynamics within this evolving digital landscape [52].

## Research methodology

### Research model

Based on the literature reviews, a conceptual model depicting the relationship between each construct is illustrated in **Fig 1**. The constructs of Performance Expectancy (PE), Effort Expectancy (EE), Social Influence (SI), Facilitating Conditions (FC), Hedonic Motivation (HM), Habit (HB), and Price Value (PV) are directly associated with the extended Unified Theory of Acceptance and Use of Technology (UTAUT2). UTAUT2, in turn, is directly associated with Consumer Engagement (CE) and Intention to Use Metaverse (IU), while CE is directly associated with IU. Finally, Social Media Marketing (SMM) is directly associated with UTAUT2, CE, and IU. This model also includes gender, age, and experience as moderators among the relations between each causal construct.

### Hypothesis development

Research has shown that UTAUT2, with its comprehensive set of constructs, directly influences both Consumer Engagement (CE) and intention to use metaverse (IU) platforms. For instance, Roy et al. [53] explored how smart retail technologies (SRT) influence customer engagement behavior by combining meta-UTAUT and SRT characteristics to analyze survey data and measure customer engagement. The results revealed the intricate relationship between SRT characteristics and customer attitudes, intentions, and engagement behavior. Despite being in its early stages in consumer markets, some innovative retailers have integrated AR into their mobile apps. For example, McLean & Wilson [54] introduced new AR attributes, including AR novelty, AR interactivity, and AR vividness, and examined their influence on perceived ease of use, usefulness, enjoyment, and subjective norms. The study found positive perceptions of AR attributes and technology acceptance attributes positively affect brand engagement through AR mobile applications, and engagement with AR-enabled brands leads to increased satisfaction with the app experience and future brand usage intention. Recently, Yang et al. [48] examined factors influencing college students' intentions to use metaverse technology, employing the UTAUT2 framework to propose a new research model. Through quantitative research with valid samples, the study highlights the significance of habits and attitudes in the success of basketball learning in a metaverse. Additionally, the study

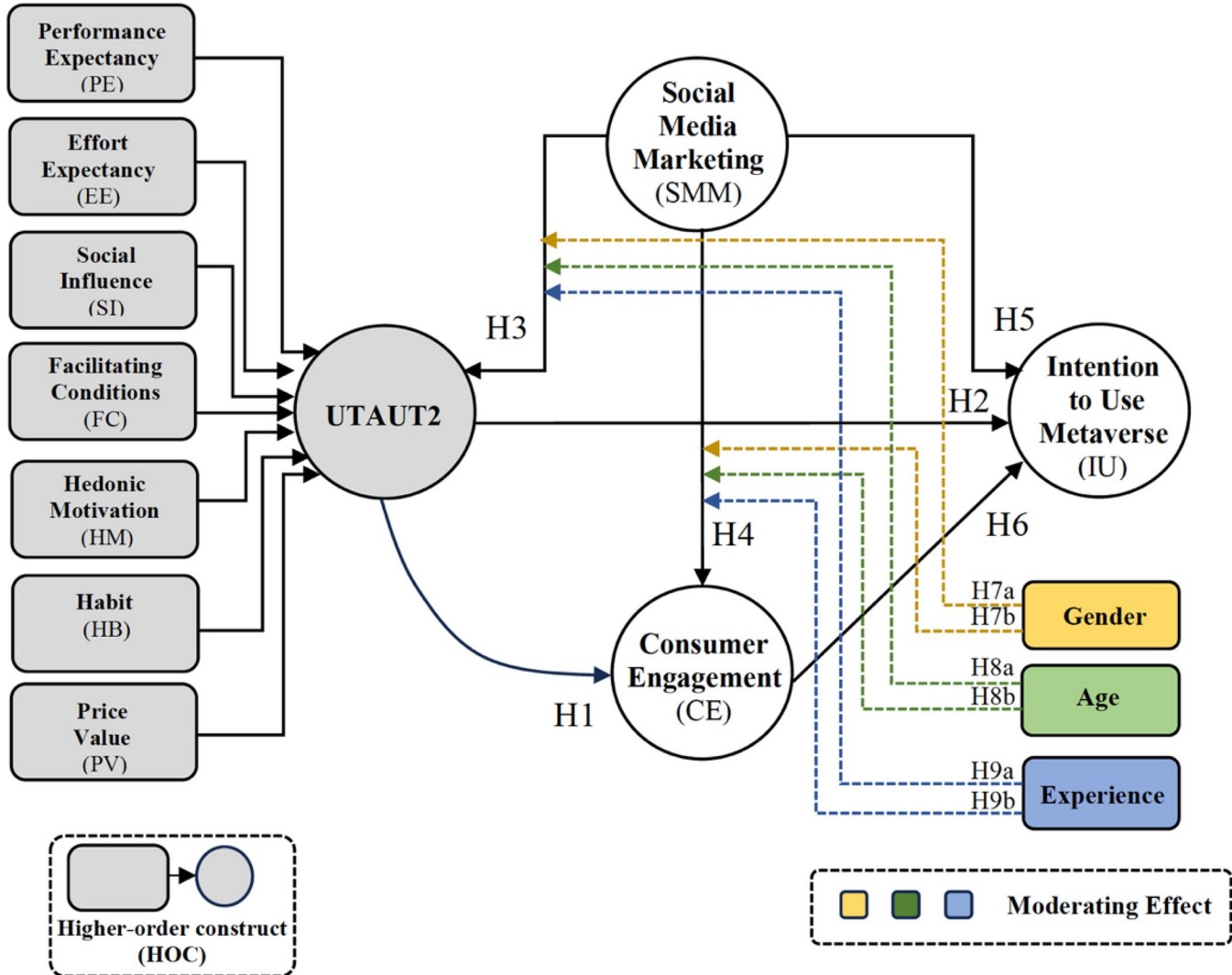

**Fig 1. Proposed conceptual model in metaverse context.**

identified grade and gender as moderator variables. Another study by Zaidan et al [55] examined how technology acceptance factors influence metaverse user behavior, with a focus on the mediation of self-efficacy. Analyzing data from 384 virtual network users using structural equation modeling, the study found that technology acceptance elements impact metaverse user behavior. Moreover, self-efficacy fully mediates the relationship between skill receipt model elements and metaverse usage. According to the aforementioned literature, the following hypotheses were formulated:

H1: UTAUT2 affects CE.

H2: UTAUT2 affects IU.

In recent years, there has been remarkable growth in social network sites. Research indicates that social media marketing activities can positively impact UTAUT2 constructs, such as Performance Expectancy (PE), Effort Expectancy (EE), Social Influence (SI), and Habit, thereby influencing customer engagement and intention to use metaverse platforms. For

instance, Herrero et al. [56] empirically tested this in the tourism industry with a sample of 537 tourists, revealing that performance expectancy, hedonic motivation, and habit are key drivers of users' intentions to share content on social networks. Ultimately, this contributes to increased customer engagement [57] and a greater willingness and intention to use them [58]. Based on the literature search, the following hypotheses were developed:

H3: SMM affects UTAUT2.

H4: SMM affects CE.

H5: SMM affects IU.

Consumer engagement plays a pivotal role in shaping individuals' intentions to use products or services. When consumers actively interact with a brand or product, they develop a sense of attachment and loyalty, which, in turn, influences their intention to continue using it. Cheung et al. [59] examined reasons for engaging with brands online, user behaviors, and loyalty, finding that most social network users followed brand fan pages despite differing motivations. The study measured users' behavioral and attitudinal loyalty and established their trust in brand information obtained through social media. Engaged consumers are more likely to advocate for the brand [60], participate in promotional activities [61], and provide valuable feedback [60], all of which contribute to a positive perception of the product or service. Additionally, engaged consumers often exhibit higher levels of satisfaction and are more inclined to recommend the brand to others [60], further reinforcing their intention to use it in the future. Thus, fostering consumer engagement is essential for building long-term relationships with customers and driving continued usage of products or services. Based on the aforementioned literature, the following hypothesis was proposed:

H6: CE affects IU.

Various studies have highlighted the crucial moderating roles of gender, age, and experience in the relationship between social media marketing, the Unified Theory of Acceptance and Use of Technology 2 (UTAUT2), and consumer engagement, influencing individuals' perceptions and behaviors in the digital realm. For instance, Zaiskaite-Jakste & Damasevicius [62] analyzed and evaluated the impact of brand-related content published on social media from the consumer perspective, focusing specifically on gender-related differences in online brand communication. The results demonstrated that women are more influenced by interactions with brand-seeking motives, which have a bigger impact on brand equity dimension awareness, while men were more influenced by hedonistic motives, affecting their engagement at the participation level.

Hazzam [63] investigated the impact of age on the relationships between informative, interactive, and trendy social media marketing activities, customer brand engagement, and brand loyalty. The results showed that the informativeness of SMM activities relates positively and significantly to customer brand engagement across all age groups. However, the strength and significance of interactive and trendy social media activities varied between age groups.

Additionally, a study by Jafar & Ahmad [64] found that experiences such as feeling immersed, escaping reality, and enjoying the metaverse had a strong positive effect on users' overall experience. Moreover, the depth of users' thoughts about their metaverse experience also influenced their satisfaction and loyalty, with the level of user involvement playing a moderating role in this relationship.

Based on the aforementioned research, we formulated our hypotheses to test as follows:

H7a: The influence of SMM on UTAUT2 is moderated by gender.

H7b: The influence of SMM on CE is moderated by gender.

H8a: The influence of SMM on UTAUT2 is moderated by age.

H8b: The influence of SMM on CE is moderated by age.

H9a: The influence of SMM on UTAUT2 is moderated by experience.

H9b: The influence of SMM on CE is moderated by experience.

## Higher-order construct of UTAUT2 in metaverse context

The UTAUT2 variables within the metaverse framework were proposed as a higher-order reflective-formative construct, comprising seven lower-order constructs measured reflectively (PE, EE, SI, FC, HM, HB, and PV). The UTAUT2 model was introduced as a second-order construct, combining reflective and formative elements. This approach is rooted in a methodological perspective, aiming to streamline the model, reduce postulated relationships, and address collinearity issues, ultimately ensuring reliable and valid empirical results [65–67]. Previous research on UTAUT2 variables in the metaverse setting has explored experience as a multifaceted concept, encompassing perceived enjoyment, perceived ease of use, social influence, facilitating conditions, hedonic motivation, habit (HB), and perceived Value (PV) [39]. This prior discourse establishes a foundation for consolidating all seven UTAUT2 dimensions in the metaverse into a unified higher-level construct, supported by independent studies highlighting the substantial impact of these constructs on the intention to use.

## Questionnaire design, research subject, and rating scales used

The quantitative study investigates factors and outcomes associated with social media marketing and the intention to use the metaverse. The questionnaire utilized in this research was adapted from existing scales, with adjustments made to align with the metaverse context, including reverse encoding to rectify response bias. Before distributing the questionnaire, three faculty members provided guidance on statement phrasing and order.

Potential respondents were initially identified by searching for hashtags related to the "metaverse" on social media platforms, such as #metaverse, #metaverseinsocialmedia, #metaversebrand, and #metaversegames. A chat box feature on platforms like Facebook Messenger and Line was employed to inform them about the purpose of contact and seek permission to gather more information about their metaverse experiences. If respondents agreed, they were provided with a hyperlink to access a Google Form divided into four sections:

**Section 1** encompassed screening questions, with respondents aged 18 years or older answering questions about metaverse engagement and familiarity with related terms. These activities included chatting with people on social media platforms, playing online games, buying online products, studying online, using smart glasses, playing games to make money or buying and selling digital assets, doing business, and traveling to various places in the virtual world.

**Section 2** gathered demographic information from participants.

**Section 3** addressed metaverse experience, covering 10 aspects such as Performance Expectancy (PE), Effort Expectancy (EE), Social Influence (SI), Facilitating Conditions (FC), Hedonic Motivation (HM), Price Value (PV), Habit (HB), Social Media Marketing (SMM), Consumer Engagement (CE), and intention to use the metaverse (IU). Questions in this section were adapted from various sources, including [29, 33, 48, 68–74], and all questions were rated on a 7-point Likert scale, ranging from strongly disagree (1) to strongly agree (5), totaling 50 questions.

**Section 4** allowed for additional written suggestions.

For participants who passed the screening questions in section 1, sections 2–4 continued, requiring approximately 20 minutes to complete. Details of all questions in sections 1–4 used in the investigation were provided in the **S1 File**.

### Ethical issues in this research with human participants

In this study, participant consent was obtained verbally in line with ethical guidelines. Prior to participation, individuals were provided with comprehensive information about the study's objectives, procedures, risks, and benefits, ensuring their ability to make informed decisions. Verbal consent was obtained from each participant, clearly communicating the study's purpose and procedures, followed by their verbal agreement to participate. Participants had the option to refuse or withdraw consent for any reason without penalty. Minors were excluded from our survey using screening questions before administering the questionnaire. These procedures were implemented to protect participants' rights and welfare, ensuring adherence to ethical standards. This study was approved by the Mahidol University Central Institutional Review Board (MU-CIRB) under Protocol Number MU-CIRB 2023/191.1506.

### Data collection

The research population for this study comprised internet users residing in the Bangkok metropolitan areas, including Bangkok and its five surrounding provinces: Nakhon Pathom, Pathum Thani, Samut Sakhon, Samut Prakan, and Nonthaburi, during August and September 2023. This geographical scope was selected due to the growing internet usage and high accessibility rates in these regions [75], indicating significant market potential for social media usage. Upon distributing the online survey link, a total of 420 responses were received. After excluding incomplete responses, the final usable sample size comprised 403 responses for subsequent statistical analysis.

The necessary sample size was determined using G*Power package version 3.1.9.4 in the linear multiple-regression analysis group, with a minimum power of 0.80, an alpha level of 0.05, a medium effect size ($f^2$) of 0.02, and 13 predictors, resulting in a sample size of 395 [76, 77]. The final sample of this study, comprising 403 responses, was deemed sufficient.

Among the 403 respondents, the majority (57.1%) identified as male, with the most prevalent age group being 22–25 years old (51.1%). A significant portion of respondents (56.8%) reported pursuing a bachelor's degree, and the most common occupation among participants was student (63%). Monthly income varied, with the largest group (37.2%) earning below 5,000 baht. Family household sizes predominantly ranged from 2 to 4 people (69.5%). Bangkok emerged as the most common province of residency among the respondents, with 28.3% residing in the capital city. Notably, the predominant means of engaging in metaverse activities among respondents was through a notebook computer. All demographic characteristics of the 403 respondents are displayed in **S1 Table**. The objective of this study was to evaluate the questionnaire and validate the suggested research model and hypotheses through a two-stage process involving Partial Least Squares Structural Equation Modeling (PLS-SEM).

### Data analysis

**Basics of data analysis.** The collected data is in the form of survey data, which may exhibit non-normal characteristics due to various factors. The main factors include inherent variability, limited dataset range, and the presence of categorical or ordinal data. The Likert scale data is predominantly ordinal and lacks a continuous distribution. It is crucial to acknowledge that not all statistical studies necessitate normal assumptions. Non-normal data are commonly

examined using non-parametric tests. Partial Least Squares Structural Equation Modeling (PLS-SEM) is a statistical modeling technique that is particularly successful in handling non-normal data. PLS-SEM is frequently utilized because of the presence of non-normal data, short sample sizes, and intricate models [78].

**Common method bias (CMB).** Given that we obtained 403 valid replies through a self-administered method, where both independent and dependent variables were measured simultaneously, a potential issue regarding common method bias (CMB) in our data arises. To address this concern, participants were assured of the anonymity and confidentiality of their responses. They were also informed that there were no unequivocally correct or incorrect answers for any item.

We employed the common method bias (CMB) test outlined by Kock [79]. This test evaluates full collinearity to ensure the dataset does not exhibit any CMB issues, with variance inflation factor (VIF) values below the threshold of 3.30. In our study, the maximum VIF value observed in the sample was 3.28, below the threshold. Additionally, we conducted Harman's single-factor test to assess the potential presence of CMB in our dataset. The unrotated single component accounted for only 40.9% of the variation, which is lower than the recommended criterion of 50%. These findings suggest that the presence of CMB is unlikely to be a major issue in our dataset [80].

**Measurement models in Partial Least Squares Structural Equation Modeling (PLS-SEM).** Due to PLS-SEM being identified as the optimal choice for data analysis, we utilized it to examine the reflective-formative second-order construct, as delineated by Hair et al. [78]. The data were analyzed using RStudio with the SEMinR package [81, 82] for Partial Least-Squares Structural Equation Modeling (PLS-SEM), a multivariate statistical analysis technique employed for hypothesis testing.

The conceptual framework adopts a reflective-formative model, evaluated against two criteria: 1) reflective measurement model evaluation (First-order reflective constructs), and 2) measurement model evaluation of higher-order constructs (HOC) [78].

1. *Assessment of measurement models (First-order reflective constructs).* The assessment of the reflective measurement model began with analyzing the indicator loadings (Bootstrap mean, β) of the constructs. First-order constructs with indicator loadings above the threshold value of 0.708 explained at least 50% of the variance in the indicators (**Table 1**), indicating item reliability [78]. According to **Table 1**, both composite reliability and Cronbach's alpha for all the measurement constructs were above the cutoff value of 0.70 [78]. The resulting Cronbach's alpha reliability values between 0.720 and 0.899 are deemed acceptable in exploratory research. The third stage focused on assessing the convergent validity of the measurement constructs using the average variance extracted (AVE) metric. All the AVE values of the constructs, as displayed in **Table 1**, exceeded the cutoff value of 0.50 [78], indicating convergent validity for all the constructs. In the fourth step of the reflective measurement model, the discriminant validity of the constructs was evaluated. According to the criterion of Fornell & Larcker [83], the shared variance between the constructs was compared with the average variance extracted from each construct. The diagonal elements in **Table 2** represent the square root of the AVE. Discriminant validity was confirmed for this measurement model, as the shared variance between the constructs was lower than the correlation coefficients (r) for each construct (**Table 2**).

2. *Measurement model evaluation of higher-order constructs (HOC).* To evaluate the higher-order UTAUT2 construct, we adopted a two-stage approach following the methodology outlined by Hair et al. [78]. Initially, we assessed the convergent and discriminant validity of all first-order constructs within UTAUT2, ensuring reliability and validity through

**Table 1. Construct reliability and validity.**

| Construct | Indicator | Mean | Median | Standard Deviation | Bootstrap Mean (β) | Bootstrap SD | Cronbach's alpha | rho_C | AVE | rho_A |
|---|---|---|---|---|---|---|---|---|---|---|
| PE | PE1 | 5.499 | 6 | 1.052 | 0.834 | 0.023 | 0.844 | 0.895 | 0.682 | 0.845 |
| | PE2 | 4.990 | 5 | 1.230 | 0.781 | 0.026 | | | | |
| | PE3 | 5.404 | 5 | 1.229 | 0.843 | 0.021 | | | | |
| | PE4 | 5.295 | 5 | 1.209 | 0.838 | 0.020 | | | | |
| EE | EE1 | 5.208 | 5 | 1.082 | 0.884 | 0.013 | 0.720 | 0.877 | 0.781 | 0.720 |
| | EE2 | 5.261 | 5 | 1.271 | 0.882 | 0.013 | | | | |
| SI | SI1 | 5.429 | 5 | 0.955 | 0.752 | 0.027 | 0.836 | 0.884 | 0.604 | 0.837 |
| | SI2 | 5.469 | 6 | 1.089 | 0.759 | 0.022 | | | | |
| | SI3 | 5.404 | 5 | 1.080 | 0.792 | 0.021 | | | | |
| | SI4 | 5.228 | 5 | 1.119 | 0.767 | 0.038 | | | | |
| | SI5 | 5.380 | 5 | 1.101 | 0.812 | 0.020 | | | | |
| FC | FC1 | 4.998 | 5 | 1.282 | 0.798 | 0.025 | 0.899 | 0.925 | 0.712 | 0.902 |
| | FC2 | 4.762 | 5 | 1.487 | 0.847 | 0.013 | | | | |
| | FC3 | 4.759 | 5 | 1.625 | 0.842 | 0.017 | | | | |
| | FC4 | 4.965 | 5 | 1.310 | 0.870 | 0.016 | | | | |
| | FC5 | 5.032 | 5 | 1.266 | 0.858 | 0.015 | | | | |
| HM | HM1 | 5.608 | 6 | 0.965 | 0.752 | 0.027 | 0.799 | 0.868 | 0.623 | 0.804 |
| | HM2 | 5.608 | 6 | 1.113 | 0.812 | 0.025 | | | | |
| | HM3 | 5.107 | 5 | 1.151 | 0.767 | 0.026 | | | | |
| | HM4 | 5.208 | 5 | 1.070 | 0.819 | 0.018 | | | | |
| HB | HB1 | 5.213 | 5 | 1.029 | 0.778 | 0.025 | 0.810 | 0.876 | 0.638 | 0.817 |
| | HB2 | 5.174 | 5 | 1.189 | 0.738 | 0.026 | | | | |
| | HB4 | 5.216 | 5 | 1.044 | 0.832 | 0.017 | | | | |
| | HB5 | 5.258 | 5 | 1.014 | 0.841 | 0.019 | | | | |
| PV | PV2 | 5.762 | 6 | 1.223 | 0.729 | 0.034 | 0.821 | 0.881 | 0.650 | 0.834 |
| | PV3 | 5.610 | 6 | 1.131 | 0.855 | 0.018 | | | | |
| | PV4 | 5.754 | 6 | 1.061 | 0.840 | 0.022 | | | | |
| | PV5 | 5.506 | 6 | 1.121 | 0.790 | 0.029 | | | | |
| SNM | SNM1 | 5.288 | 5 | 0.899 | 0.728 | 0.028 | 0.830 | 0.881 | 0.598 | 0.833 |
| | SNM2 | 5.270 | 5 | 1.146 | 0.703 | 0.026 | | | | |
| | SNM3 | 5.352 | 5 | 1.129 | 0.800 | 0.021 | | | | |
| | SNM4 | 5.362 | 5 | 1.040 | 0.836 | 0.017 | | | | |
| | SNM5 | 5.442 | 5 | 1.053 | 0.786 | 0.021 | | | | |
| CE | CE1 | 5.305 | 5 | 0.992 | 0.738 | 0.034 | 0.824 | 0.877 | 0.588 | 0.831 |
| | CE2 | 5.144 | 5 | 1.201 | 0.703 | 0.031 | | | | |
| | CE3 | 5.002 | 5 | 1.294 | 0.744 | 0.035 | | | | |
| | CE4 | 4.993 | 5 | 1.265 | 0.810 | 0.020 | | | | |
| | CE5 | 5.072 | 5 | 1.181 | 0.824 | 0.020 | | | | |
| IU | BI1 | 5.196 | 5 | 0.987 | 0.806 | 0.023 | 0.862 | 0.901 | 0.645 | 0.862 |
| | BI2 | 5.094 | 5 | 1.246 | 0.747 | 0.026 | | | | |
| | BI3 | 5.370 | 5 | 1.104 | 0.800 | 0.023 | | | | |
| | BI4 | 5.256 | 5 | 1.033 | 0.821 | 0.019 | | | | |
| | BI5 | 5.392 | 5 | 1.041 | 0.829 | 0.021 | | | | |

**Notes**: AVE is average variance extracted; CR is composite reliability.

**Table 2. Discriminant validity of first-order reflective constructs using the Fornell-Larcker criterion [83].**

|  | PE | EE | SI | FC | HM | HB | PV | SMM | CE | IU |
|---|---|---|---|---|---|---|---|---|---|---|
| PE | *0.826* |  |  |  |  |  |  |  |  |  |
| EE | 0.692 | *0.884* |  |  |  |  |  |  |  |  |
| SI | 0.541 | 0.654 | *0.777* |  |  |  |  |  |  |  |
| FC | 0.562 | 0.681 | 0.622 | *0.844* |  |  |  |  |  |  |
| HM | 0.633 | 0.683 | 0.724 | 0.705 | *0.789* |  |  |  |  |  |
| HB | 0.537 | 0.644 | 0.636 | 0.666 | 0.707 | *0.799* |  |  |  |  |
| PV | 0.123 | 0.195 | 0.379 | 0.079 | 0.287 | 0.353 | *0.807* |  |  |  |
| SMM | 0.470 | 0.588 | 0.661 | 0.589 | 0.668 | 0.737 | 0.471 | *0.773* |  |  |
| CE | 0.549 | 0.678 | 0.596 | 0.676 | 0.671 | 0.653 | 0.184 | 0.676 | *0.767* |  |
| IU | 0.624 | 0.678 | 0.669 | 0.642 | 0.735 | 0.657 | 0.332 | 0.692 | 0.753 | *0.803* |

**Notes**: The square root of the AVE is italicized on the diagonal, while the correlation coefficients (r) between paired constructs are listed off the diagonal.

standard procedures. Subsequently, we analyzed UTAUT2 as a formative construct. This involved three key steps: i) Conducting redundancy analysis to identify a single global item capturing the essence of UTAUT2, aiming for specific path coefficients and R-squared values. ii) Performing a multicollinearity check using variance inflation factor (VIF) values to ensure no multicollinearity among first-order constructs (S2 Table), and iii) Verifying outer weight significance to confirm convergent validity before testing the structural model, which included the reflective-formative higher-order construct UTAUT2.

The discriminant validity results presented in **Table 3** indicate that each construct had a value below the threshold of 0.85 [84] and 0.90 [78, 85, 86], demonstrating adherence to the HTMT ratio.

UTAUT2 is specified using a repeated-indicators approach, recognized as one of the prominent methods for specifying higher-order constructs [87]. It comprises a second-order construct encompassing seven dimensions: PE, EE, SI, FC, HM, HB, and PV. These seven lower-order constructs measure the second-order construct UTAUT2, utilizing the 28 indicators (S2 Table). Evaluating the measurement quality of HOC includes three criteria i.e., convergent validity, indicator's collinearity, and indicator weights' significance and relevance.

**A. Convergent validity assessment.** To evaluate convergent validity for formative constructs, several methods are recommended by Hair et al. [78]. Two primary approaches

**Table 3. Discriminant validity of the reflective constructs using the HTMT criterion.**

|  | SMM | CE | IU | PE | EE | SI | FC | HM | HB |
|---|---|---|---|---|---|---|---|---|---|
| SMM |  |  |  |  |  |  |  |  |  |
| CE | 0.810 |  |  |  |  |  |  |  |  |
| IU | 0.818 | 0.887 |  |  |  |  |  |  |  |
| PE | 0.559 | 0.656 | 0.731 |  |  |  |  |  |  |
| EE | 0.760 | 0.873 | 0.860 | 0.888 |  |  |  |  |  |
| SI | 0.794 | 0.710 | 0.787 | 0.643 | 0.842 |  |  |  |  |
| FC | 0.678 | 0.777 | 0.726 | 0.642 | 0.843 | 0.714 |  |  |  |
| HM | 0.811 | 0.816 | 0.879 | 0.760 | 0.899 | 0.884 | 0.827 |  |  |
| HB | 0.895 | 0.792 | 0.785 | 0.648 | 0.843 | 0.770 | 0.774 | 0.871 |  |
| PV | 0.565 | 0.219 | 0.389 | 0.160 | 0.248 | 0.449 | 0.122 | 0.338 | 0.428 |

include correlating with reflective items and employing a global item. Correlating with reflective items involves calculating the correlation between the formative construct and its associated reflective measures. Acceptable values include a minimum path coefficient of 0.70 and an $R^2$ of 0.50 for the endogenous variable. Alternatively, utilizing the global item method aims to minimize respondent fatigue and improve response rates. Before integration into the final questionnaire, this item underwent pretesting. Analysis revealed a robust path coefficient of 0.809 between latent variables and an $R^2$ of 0.991 for the endogenous variable (**S2 Table**), indicating commendable convergent validity for the formative model. Furthermore, the average variance extracted (AVE) for the higher-order construct exceeded the recommended threshold of 0.50 (**S2 Table**), reinforcing its convergent validity as outlined by Arya et al. [67] and Hair et al. [78, 86]. Additionally, both the composite reliability (CR) and Cronbach's Alpha for UTAUT2 surpassed the 0.70 threshold [78, 86], as illustrated in **S2 Table**, further affirming its reliability.

**B. Indicator collinearity assessment.** In formative measurement models, unlike reflective ones, high correlations between items are not usually expected; however, such correlations can indicate problematic collinearity, as noted by Hair et al. [78, 86]. To assess potential collinearity among the formative indicators of the latent variables in UTAUT2, we utilized the variance inflation factor (VIF), focusing on the inner VIF values relevant to the second-order reflective-formative structure. Diamantopoulos & Siguaw [88] propose a VIF threshold of less than 3.3. Fortunately, as shown in **S2 Table**, all predictor latent variables exhibit VIF values below this threshold. This assures us that collinearity isn't an issue among the formative items of UTAUT2's latent variables, as emphasized by Hair et al. [78, 86].

**C. Assessment of indicator weights' significance and relevance.** Bootstrapping is a method that enables us to determine the relative importance of each indicator in the formative model, as represented by its bootstrap weight, contrasting with loadings that measure the absolute importance of each indicator. To conduct bootstrapping, we require a similar number of resamples as the original sample size, as highlighted by Hair et al. [78]. In this instance, we utilized 1000 resamples, following the suggestion by Chin [89], to assess the significance of individual indicator weights. As per Lohmöller [90], an indicator weight exceeding 0.1 signifies its significance. Our findings revealed that all formative indicators surpassed this threshold, indicating empirical support for retaining them all in the model, as emphasized by Hair et al. [78]. Additionally, the significant t-values for all indicators further corroborate their importance in the model [78].

## Results

### Structural model analysis and hypotheses testing

The findings from this study offer valuable insights into the relationships between key constructs of UTAUT2 in the context of metaverse usage. After confirming the measurement model's quality, we assessed the structural model using non-parametric bootstrapping with 5,000 samples in PLS-SEM. Prior to analyzing path relationships, we checked for collinearity using VIF values, all of which were below 5, indicating no concerns (**S2 Table**). The structural model and its path coefficients are showcased in **Fig 2** and **Table 4**, respectively. Among the hypothesized relationships (H1, H2, H3, H4, and H6), all were significant except H5. Notably, all constructs except SMM significantly influenced IU. The bootstrap mean (β) coefficients illustrated the independent variable's influence on the dependent variable, with SMM exerting the strongest influence on UTAUT2 (β = 0.787), followed by UTAUT2 on CE (β = 0.589), UTAUT2 on IU (β = 0.468), CE on IU (β = 0.333), and SMM on CE (β = 0.211), respectively.

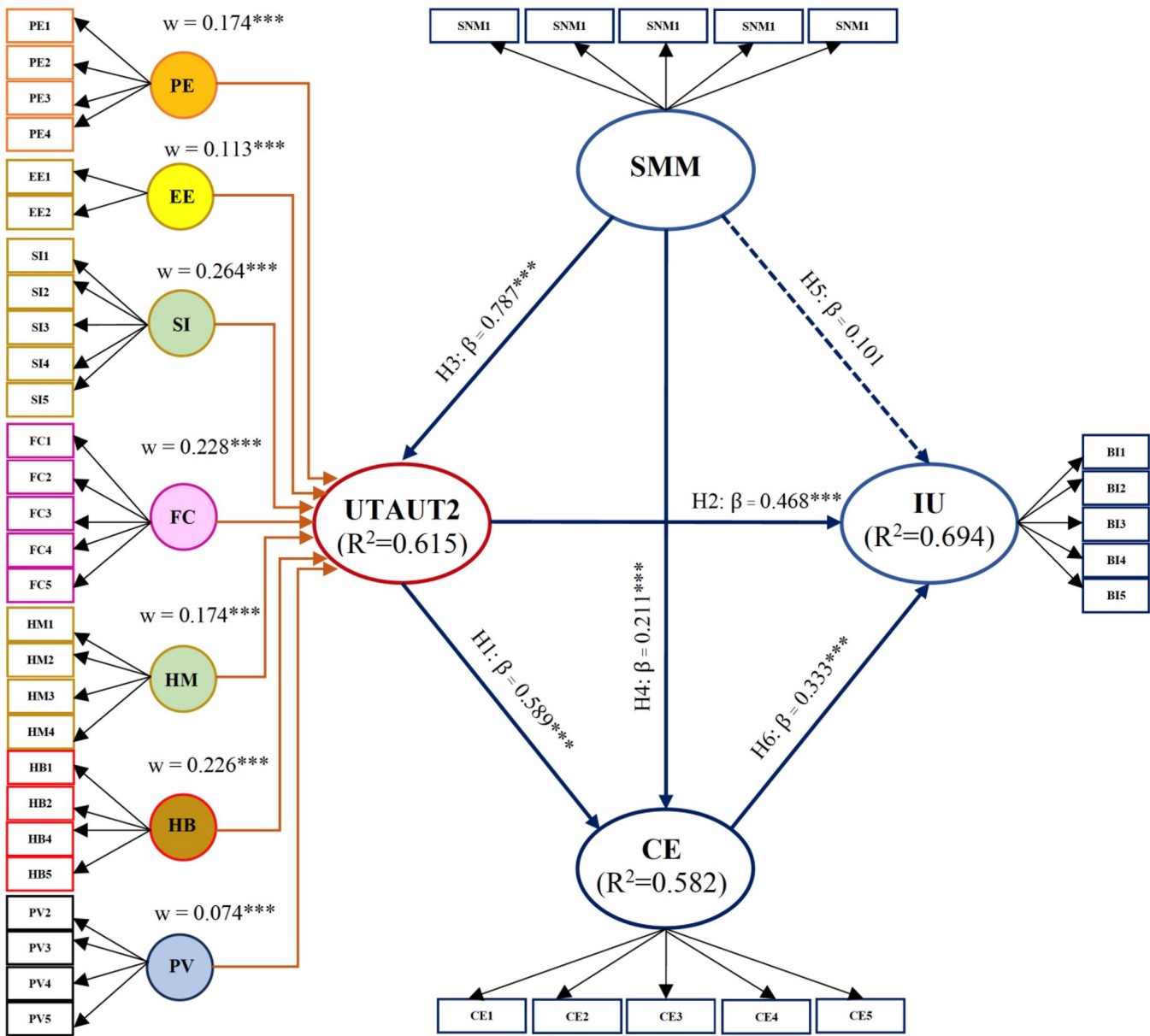

**Notes**. *p < 0.05, **p < 0.01, ***p < 0.001; w = Bootstrap weight; β = Bootstrap mean

**Fig 2. Structural model analysis and hypotheses testing.**

**Table 4. Path coefficients (direct effect).**

| Hypothesis | Path | Bootstrap Mean (β) | Bootstrap SD | T Stat. | [LLCI, ULCI] | $f^2$ value | Result |
|---|---|---|---|---|---|---|---|
| H1 | UTAUT2 -> CE | 0.589 | 0.070 | 8.270 | [0.451; 0.720] | 0.301 | Supported |
| H2 | UTAUT2 -> IU | 0.468 | 0.091 | 5.177 | [0300; 0.646] | 0.209 | Supported |
| H3 | SMM -> UTAUT2 | 0.787 | 0.022 | 35.497 | [0.741; 0.827] | 1.597 | Supported |
| H4 | SMM -> CE | 0.211 | 0.078 | 2.837 | [0.063; 0.362] | 0.046 | Supported |
| H5 | SMM -> IU | 0.101 | 0.061 | 1.669 | [-0.019; 0.218] | 0.013 | Not supported |
| H6 | CE -> IU | 0.333 | 0.075 | 4.448 | [0.181; 0.471] | 0.151 | Supported |

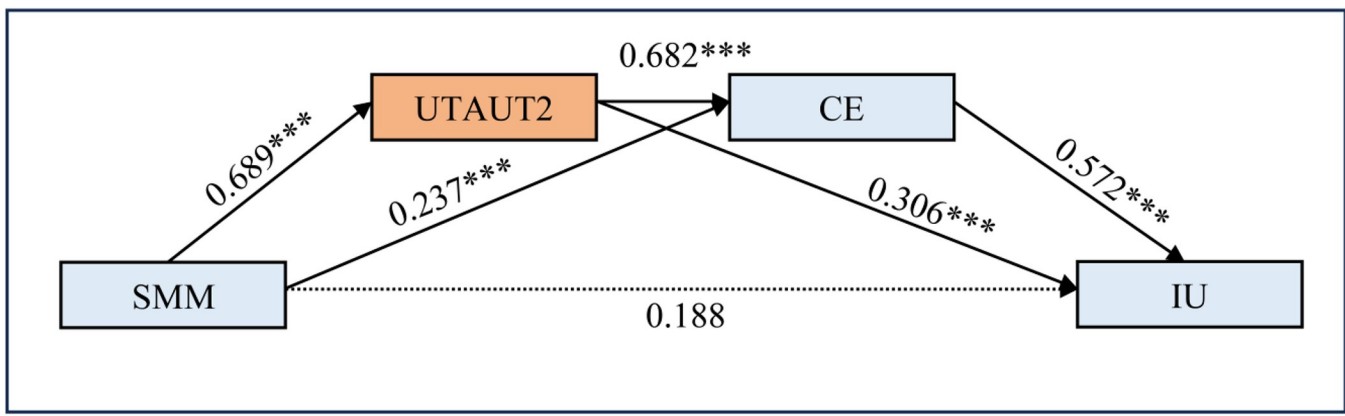

**Fig 3. Serial mediation model results.** Notes. *p < 0.05, **p < 0.01, ***p < 0.001.

Regarding predictive power, $R^2$ values indicated substantial predictive power for IU ($R^2$ = 0.694, adjusted $R^2$ = 0.692), and moderate predictive power for CE ($R^2$ = 0.582, adjusted $R^2$ = 0.580) and UTAUT2 ($R^2$ = 0.615, adjusted $R^2$ = 0.614). Effect size analysis, employing Cohen's [91] classifications, revealed a small effect for H4 ($f^2$ = 0.046), medium effects for H6, H2, and H1 ($f^2$ = 0.151, 0.209, and 0.301, respectively), and a large effect for H3 ($f^2$ = 1.597), with one hypothesis showing no effect for H5 ($f^2$ < 0.02).

## Serial mediation analysis

Mediation analysis explores how an independent variable X influences a dependent variable Y through one or more mediator variables [92]. Process 4.0 (Model 6) is suitable for this testing. Results from multiple regression analysis are presented in **Fig 3** and **Table 5**. The overall

**Table 5. Hayes' PROCESS MACRO model results of linear regression analysis.**

| Model 6 Serial mediation | | β | SE | t | p | LLCI | ULCI | $R^2$ | F |
|---|---|---|---|---|---|---|---|---|---|
| **Model 1: SMM -> IU** | | | | | | | | 0.482 | 240.676*** |
| | Constant | 1.307 | 0.264 | 4.955 | 0.000 | 0.788 | 1.825 | | |
| | SMM | 0.740 | 0.048 | 15.514 | 0.000 | 0.646 | 0.834 | | |
| **Model 2: SMM -> UTAUT2** | | | | | | | | 0.595 | 292.517*** |
| | Constant | 1.607 | 0.222 | 7.244 | 0.000 | 1.171 | 2.043 | | |
| | SMM | 0.689 | 0.040 | 17.103 | 0.000 | 0.610 | 0.769 | | |
| **Model 3: SMM -> CE; UTAUT2 -> CE** | | | | | | | | 0.561 | 154.463*** |
| | Constant | 0.034 | 0.295 | 0.115 | 0.908 | -0.546 | 0.613 | | |
| | SMM | 0.273 | 0.095 | 2.891 | 0.004 | 0.087 | 0.459 | | |
| | UTAUT2 | 0.682 | 0.098 | 6.949 | 0.000 | 0.489 | 0.875 | | |
| **Model 4: SMM -> IU; CE -> IU; UTAUT2 -> IU** | | | | | | | | 0.702 | 273.960*** |
| | Constant | 0.041 | 0.235 | 0.174 | 0.862 | -0.421 | 0.503 | | |
| | SMM | 0.118 | 0.069 | 1.704 | 0.089 | -0.018 | 0.254 | | |
| | CE | 0.572 | 0.138 | 4.147 | 0.000 | 0.301 | 0.844 | | |
| | UTAUT2 | 0.306 | 0.075 | 4.086 | 0.000 | 0.159 | 0.454 | | |

**Notes**. β = Direct effect; SE = Standard errors; LLCI = Lower-level confidence interval; ULCI = Upper-level confidence interval. *p < 0.05, **p < 0.01, ***p < 0.001.

model, encompassing SMM, CE, and UTAUT2 constructs, accounts for 70.2% of IU variance ($R^2 = 0.702$, F = 273.960). This model demonstrates a moderate predictive power [78] across all focal predictors.

SMM serves as a significant positive predictor of IU (β = 0.740; t = 15.514; p < 0.001) (**Table 5**). Specifically, SMM significantly influences UTAUT2 (β = 0.689; t = 17.103; p < 0.001) and CE (β = 0.273; t = 2.890; p < 0.001). UTAUT2 (β = 0.682; t = 6.949; p < 0.001) significantly affects CE. In predicting IU, two predictors—CE (β = 0.572; t = 4.147; p < 0.001) and UTAUT2 (β = 0.306; t = 4.086; p < 0.001)—are positively significant. However, SMM (β = 0.118; t = 1.704; p = 0.089) is not significant, indicating complete or full mediation.

## Moderated mediation process

Moderation analysis, a component of conditional process analysis, explores how the impact of variable X on consequent Y is influenced by a third variable or set of variables [93]. As hypothesized, this study expected gender, age, and technology experience to moderate the relationship between SMM and IU. Process 4.0 (Model 84) is employed for testing [93], with results presented in **Fig 4** and **S3 Table**.

**Table 6** reveals a significant change in $R^2$ ($\Delta R^2 = 0.019$, p = 0.000), indicating that age moderates the effect between SMM and UTAUT2, with an interaction coefficient of 0.057 (p < 0.05). This supports H8a, suggesting that age influences the positive relationship between SMM and UTAUT2. **Fig 4** illustrates a simple slope test, predicting UTAUT2 by SMM across varying age levels: low (-1SD), medium (average), and high (+1SD). Notably, the lines representing these levels are not parallel, signifying significant differences in slopes. The interpretations are as follows:

1. UTAUT2 level increases with higher SMM levels, particularly at higher age levels.

2. Users with higher age levels tend to express higher UTAUT2 levels.

3. Users with high SMM levels exhibit higher UTAUT2 levels compared to those with low SMM levels, and this positive impact intensifies with increasing age.

Similarly, **Table 6** also demonstrates a significant change in $R^2$ ($\Delta R^2 = 0.005$, p = 0.021), indicating that experience moderates the effect between SMM and UTAUT2, with an interaction coefficient of -0.022 (p < 0.05). This supports H9a, suggesting that experience influences the positive relationship between SMM and UTAUT2. **Fig 4** displays a simple slope test, predicting UTAUT2 by SMM across varying experience levels: low (-1SD), medium (average), and high (+1SD). Again, the lines are not parallel, indicating different slopes. The interpretations are:

1. UTAUT2 level increases with higher SMM levels, particularly with increased experience.

2. Users with higher experience levels tend to express higher UTAUT2 levels.

3. Users with high SMM levels exhibit higher UTAUT2 levels compared to those with low SMM levels, and this positive impact becomes more pronounced with increased experience.

Furthermore, the total indirect association between SMM and IU, mediated through UTAUT2 and CE, is moderated by age and experience, with moderated mediation indices of 0.012 and -0.004, respectively (Bootstrap SE = 0.004 and 0.002, Bootstrap 95% CI = [0.005, 0.021] and [-0.010, 0.000], respectively; **Table 7**).

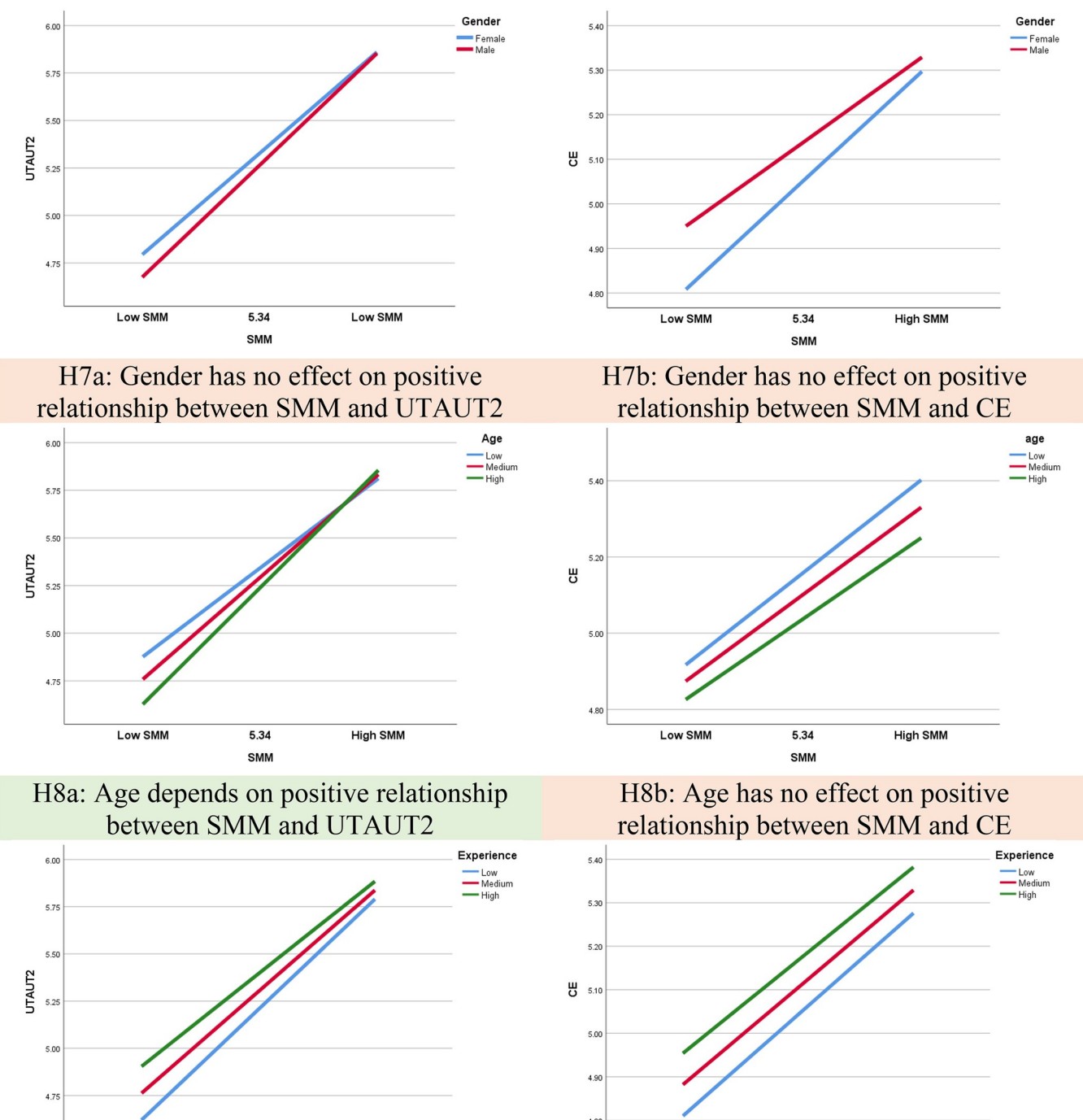

**Fig 4. Moderation effects.**

**Table 6. The output of moderated mediation analysis.**

| Predictor | Model 1 Mediator: UTAUT2 | | | Model 2 Mediator: CE | | | Model 3 Mediator: IU | | |
|---|---|---|---|---|---|---|---|---|---|
| | β | SE | t | β | SE | t | β | SE | t |
| **Gender** | | | | | | | | | |
| Constant | 1.826*** [1.407,2.245] | 0.213 | 8.560 | -0.214 [-0.810, 0.282] | 0.303 | -0.706 | 0.041[-0.309, 0.390] | 0.178 | 0.230 |
| SMM | 0.655*** [0.578, 0.732] | 0.039 | 16.755 | 0.301*** [0.169, 0.432] | 0.067 | 4.506 | 0.118*[0.025, 0.210] | 0.047 | 2.504 |
| UTAUT2 | | | | 0.692*** [0.563, 0.821] | 0.065 | 10.577 | 0.572***[0.459, 0.686] | 0.058 | 9.897 |
| Gender | -0.431 [-1.035, 0.173] | 0.307 | -1.402 | 0.447 [-0.345, 1.239] | 0.403 | 1.110 | | | |
| SMM*Gender | 0.069 [-0.043, 0.181] | 0.057 | 1.212 | -0.067 [-0.214, 0.079] | 0.074 | -0.905 | | | |
| CE | | | | | | | 0.306***[0.229, 0.384] | 0.039 | 7.796 |
| $R^2$ | 0.598*** F (198.057) | | | 0.564*** F (128.759) | | | 0.702*** F (312.662) | | |
| $\Delta R^2$ | | | | 0.001 (F = 1.469) | | | | | |
| **Age** | | | | | | | | | |
| Constant | 2.606*** [2.130, 3.082] | 0.242 | 10.765 | -0.004 [-0.735, 0.727] | 0.372 | -0.011 | 0.041[-0.309, 0.390] | 0.178 | 0.230 |
| SMM | 0.519*** [0.429, 0.608] | 0.046 | 11.396 | 0.310*** [0.171, 0.450] | 0.071 | 4.379 | 0.118*[0.025, 0.210] | 0.047 | 2.504 |
| UTAUT2 | | | | 0.670*** [0.537, 0.803] | 0.068 | 9.894 | 0.572***[0.459, 0.686] | 0.058 | 9.897 |
| Age | -0.337*** [-0.466, -0.208] | 0.066 | -5.141 | 0.025 [-0.155, 0.205] | 0.092 | 0.271 | | | |
| SMM*Age | 0.057*** [-0.466, -0.208] | 0.013 | 4.494 | -0.012 [-0.046, 0.023] | 0.018 | -0.675 | | | |
| CE | | | | | | | 0.306***[0.229, 0.384] | 0.039 | 7.796 |
| $R^2$ | 0.626*** F (222.796) | | | 0.566*** F (544.000) | | | 0.702*** F (312.662) | | |
| $\Delta R^2$ | | | | 0.019*** F (20.199) | | | | | |
| **Experience** | | | | | | | | | |
| Constant | 0.483 [-0.393, 1.359] | 0.445 | 1.084 | -0.226 [-1.395, 0.943] | 0.595 | | 0.041[-0.309, 0.390] | 0.178 | 0.230 |
| SMM | 0.846*** [0.683, 1.008] | 0.083 | 10.212 | 0.312* [0.069, 0.556] | 0.124 | | 0.118*[0.025, 0.210] | 0.047 | 2.504 |
| UTAUT2 | | | | 0.655*** [0.524, 0.786] | 0.067 | | 0.572***[0.459, 0.686] | 0.058 | 9.897 |
| Experience | 0.153** [0.050, 0.256] | 0.052 | 2.929 | 0.047 [-0.091, 0.186] | 0.070 | | | | |
| SMM*Experience | -0.022*[-0.041, -0.003] | 0.010 | -2.314 | -0.004 [-0.030, 0.021] | 0.013 | | | | |
| CE | | | | | | | 0.306***[0.229, 0.384] | 0.039 | 7.796 |
| $R^2$ | 0.615*** F (212.159) | | | 0.565*** F (129.383) | | | 0.702*** F (312.662) | | |
| $\Delta R^2$ | | | | 0.005* F (5.355) | | | | | |

Notes

*$p < 0.05$

**$p < 0.01$

***$p < 0.001$.

## Artificial neural network (ANN) model

This study employs both Partial Least Squares Structural Equation Modeling (PLS-SEM) and artificial neural networks (ANN) (Fig 2) to analyze data. PLS-SEM is traditionally used for exploring linear relationships between constructs, while ANN is preferred for identifying non-linear connections among variables [94]. Although SEM is commonly used in studying causal relationships, its integration with ANN is limited [95]. SEM's limitation lies in its ability to examine only linear relationships, potentially oversimplifying human decision-making processes. Therefore, combining SEM with ANN can overcome SEM's weaknesses and enhance the analysis.

The study utilized IBM SPSS version 21 for ANN analysis, employing a feed-forward back-propagation multilayer training technique with a hyperbolic tangent function [96]. SMM, UTAUT2, and CE from Fig 5 constituted the ANN model's input layer, while IU served as the

**Table 7. Index of moderated mediation effects on IU.**

| Moderator | Moderator value (Indirect effect of SMM->UTAUT2->CE->IU) | | | |
|---|---|---|---|---|
| **Gender** | **Conditional indirect effect** | | | |
| | **Effect** | **Boot SE** | **Boot LLCI** | **Boot ULCI** |
| Female | 0.139 | 0.046 | 0.063 | 0.245 |
| Male | 0.153 | 0.046 | 0.074 | 0.249 |
| | **Conditional total indirect effect** | | | |
| | **Index** | **Boot SE** | **Boot LLCI** | **Boot ULCI** |
| SMM*Gender | 0.015 | 0.016 | -0.019 | 0.046 |
| **Age** | **Conditional indirect effect** | | | |
| | **Effect** | **Boot SE** | **Boot LLCI** | **Boot ULCI** |
| Low Age, -1 SD | 0.118 | 0.038 | 0.056 | 0.205 |
| Medium | 0.136 | 0.042 | 0.065 | 0.229 |
| High Age, +1 SD | 0.155 | 0.047 | 0.075 | 0.259 |
| | **Conditional total indirect effect** | | | |
| | **Index** | **Boot SE** | **Boot LLCI** | **Boot ULCI** |
| SMM*Age | 0.012 | 0.004 | 0.005 | 0.021 |
| **Experience** | **Conditional indirect effect** | | | |
| | **Effect** | **Boot SE** | **Boot LLCI** | **Boot ULCI** |
| Low Experience, -1 SD | 0.144 | 0.045 | 0.069 | 0.244 |
| Medium | 0.133 | 0.042 | 0.064 | 0.226 |
| High Experience, +1 SD | 0.121 | 0.039 | 0.057 | 0.208 |
| | **Conditional total indirect effect** | | | |
| | **Index** | **Boot SE** | **Boot LLCI** | **Boot ULCI** |
| SMM*Experience | -0.004 | 0.002 | -0.010 | 0.000 |

**Notes**. Boot SE = Bootstrap standard error; Boot LLCI = Bootstrap lower level confidence interval; Boot ULCI = Bootstrap upper level confidence interval.

output layer. The model's accuracy was evaluated using the root mean square error (RMSE) from ten networks, with 30% of the data for testing and 70% for training, along with ten-fold cross-validation to prevent overfitting [97] (**S3 Table**). The study reported mean RMSE values of 0.3854 for training and 0.3910 for testing, indicating high accuracy of the ANN models [94].

Overall, the integration of PLS-SEM and ANN represents a methodological advancement in data analysis, allowing for a more comprehensive understanding of the intricate relationships within the research domain. By leveraging the complementary strengths of these methodologies, the study not only overcomes the limitations of traditional SEM but also enhances the validity and robustness of the analysis.

The sensitivity analysis conducted within the artificial neural network (ANN) framework sheds light on the relative importance of exogenous constructs in predicting endogenous constructs. This analysis provides valuable insights into the drivers of the phenomenon under investigation and helps prioritize resources and interventions accordingly. The normalized importance (NI) of each independent input variable, namely SMM, UTAUT2, and CE, was calculated and ranked. These rankings offer a quantitative understanding of the impact of each variable on the endogenous construct, IU. The results of the sensitivity analysis revealed that UTAUT2 emerged as the most significant predictor, with a normalized importance of 100%, followed by CE (77%). Interestingly, SMM exhibited the least impact, accounting for only 28% of the normalized importance in IU (**S4 Table**). These findings highlight the differential influence of each exogenous construct on the outcome variable and underscore the importance of

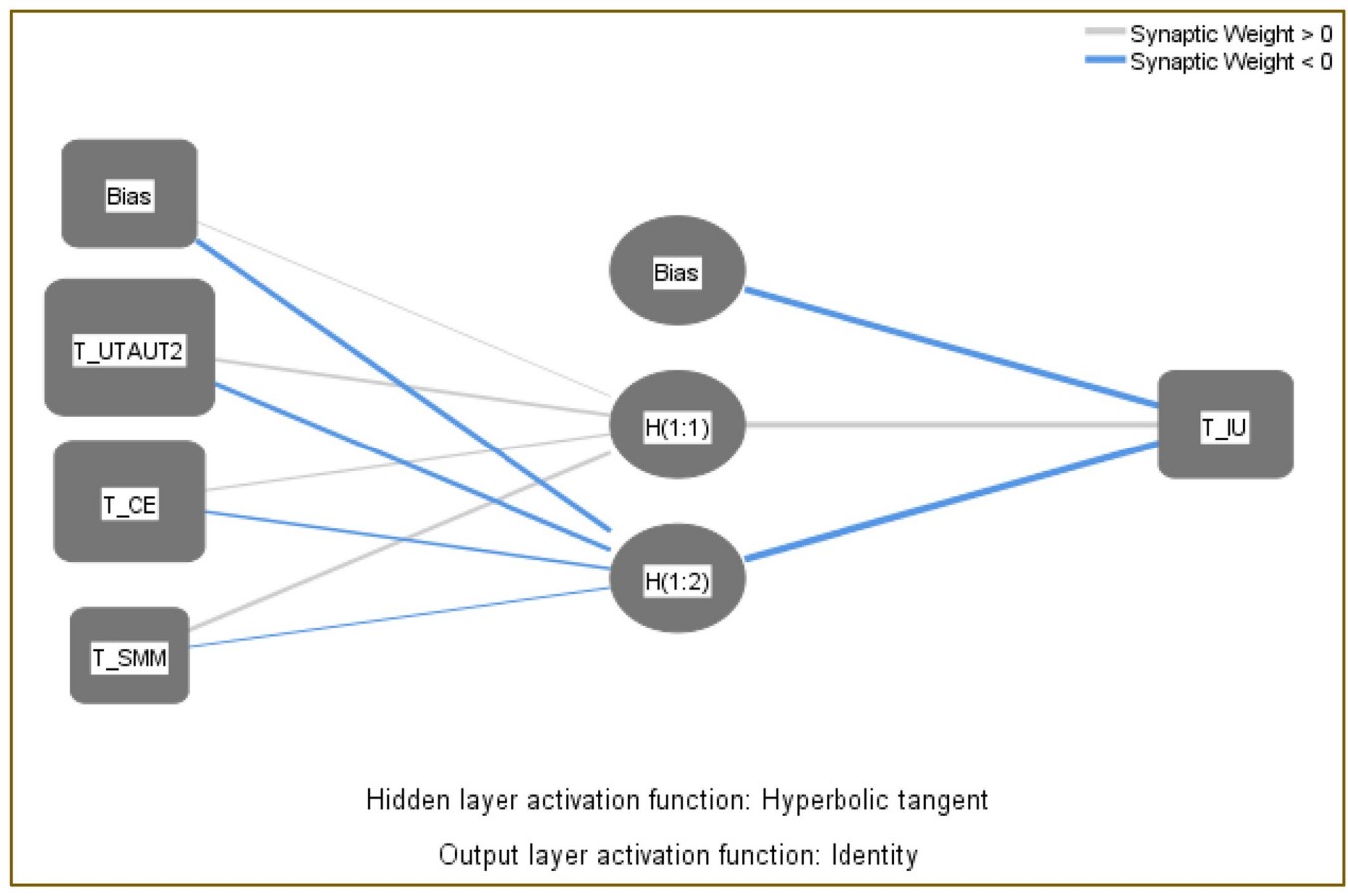

**Fig 5. ANN model.**

considering multiple factors in predictive modeling. Finally, **Table 8** displays the rankings of SMM, UTAUT2, and CE, where UTAUT2 ranked first in both PLS-SEM and ANN analyses.

## Discussion

### Demographic insights into metaverse adoption among residents of Bangkok metropolitan areas

The demographic profile of the 403 respondents offers valuable insights into how Thais living in Bangkok metropolitan areas are adopting metaverse technology. A significant majority of respondents were male, and the most common age group was 22–25 years old, indicating potential preferences based on gender and age in metaverse adoption. Moreover, a substantial number of respondents were pursuing bachelor's degrees, suggesting a higher educational level among users. The predominant occupation was student, highlighting the active

**Table 8. Comparison between PLS-SEM and ANN analysis (output: IU).**

| Predictor | Path | PLS-SEM rank | ANN-normalized relative important (%) | ANN rank | Congruence |
|-----------|------|--------------|----------------------------------------|----------|------------|
| SMM | 0.101 | 3 | 28 | 3 | Yes |
| UTAUT2 | 0.468 | 1 | 100 | 1 | Yes |
| CE | 0.333 | 2 | 77 | 2 | Yes |

engagement of younger individuals, particularly students, with metaverse platforms. Despite diverse educational backgrounds, respondents' monthly incomes varied, with a significant portion earning below 5,000 baht, indicating accessibility across different income levels. Family household sizes mainly ranged from 2 to 4 people, indicating potential family dynamics influencing metaverse usage patterns. Bangkok emerged as the most common province of residency, indicating a concentration of metaverse users in urban areas. Notably, Thailand's position as one of Southeast Asia's frontrunners in ecommerce and 5G technologies, with its digitally literate population playing an important role in the metaverse gaming industry [98], underscores the significance of these demographic trends. The primary means of engaging in metaverse activities among respondents was through a notebook computer, emphasizing the importance of personal computing devices. These demographic insights provide a comprehensive understanding of metaverse usage patterns among Thais in Bangkok metropolitan areas and offer valuable implications for future research and marketing strategies in the metaverse space.

## Direct effects

The relationship among intention to use technology, consumer engagement, social media marketing, and the Unified Theory of Acceptance and Use of Technology 2 (UTAUT2) is a multifaceted interplay that has been extensively examined across various contexts. UTAUT2, an extension of the original UTAUT model, introduces additional constructs such as hedonic motivation, habit, and price value, which impact behavioral intention and technology use [99]. Research has demonstrated that UTAUT2 significantly enhances our understanding of behavioral intention and technology use compared to the original UTAUT model [100]. In this study, we leverage the UTAUT2 framework to explore the motivations of Thai users in adopting innovative digital platforms such as metaverse technology. Our findings revealed that all constructs, including consumer engagement and UTAUT2 (excluding social media marketing), significantly influence the intention to use metaverse technology among Thai users (IU). This underscores the intricate nature of the factors influencing the adoption and utilization of metaverse technology.

The findings confirm that UTAUT2 directly influences consumer engagement (H1) and intention to use metaverse technology (H2) among Thai users. Consistent with prior research, this study illustrates that UTAUT2 constructs exert direct effects on consumers' engagement levels with technological use and their intention to adopt new technology. For instance, Kwateng et al. [101] investigated the acceptance and usage of mobile banking using the UTAUT2 model, highlighting the impact of constructs such as habit, price value, and trust on behavioral intention and actual technology utilization in Ghana. Similarly, in healthcare contexts, Tavares & Oliveira [102] demonstrated the relevance of UTAUT2 in understanding consumer behavior regarding the adoption of electronic health record patient portals among citizens from the USA, Canada, and Bangladesh. Additionally, Baptista & Oliveira [103] delved into the influence of gamification on the acceptance of mobile banking services within the UTAUT2 framework. This study expanded the UTAUT2 model to incorporate gamification effects, uncovering how interactive elements can augment consumer engagement with technology. The consideration of gamification within the UTAUT2 framework aids researchers in gaining deeper insights into designing engaging technology solutions that foster consumer adoption.

The study highlights the pivotal role of social media marketing in driving UTAUT2 (H3) and influencing consumer engagement (H4). Various studies offer valuable insights into the impact of social media marketing on UTAUT2 and consumer engagement. For instance, Gharaibeh et al. [104] discovered that social media, along with social influence, effort expectancy, hedonic motivation, performance expectancy, habit, and facilitating conditions,

significantly affect Jordanian consumers' intention to adopt mobile commerce. Devereux et al. [105] analyzed social media posts (Facebook, Instagram, and Twitter) from small retail firms (N = 2,607) and demonstrated how small retailers can enhance consumer engagement through social media marketing strategies. Bright et al. [106] examined social media fatigue and its antecedents, including social media efficacy, helpfulness, confidence, and privacy concerns. Their study highlighted that factors such as social media efficacy and privacy concerns could impact consumer engagement. Privacy concerns and confidence emerged as the most significant predictors of social media fatigue, leading users to withdraw from social media participation due to feeling overwhelmed by information. Additionally, Meire et al. [107] explored how firms' social media engagement initiatives during customer experiential events influence customer sentiment in digital engagement. Their results revealed that marketers can influence customer sentiment in digital engagement beyond performance during interactions. Informational marketer-generated content is more effective than emotional content in enhancing customer sentiment for unfavorable event outcomes. This study underscores how firms' social media initiatives shape customer sentiment and digital engagement.

Although metaverse technology is poised to disrupt current lifestyles and bring about significant changes in various sectors such as the economy, society, politics, and business [108], the study revealed that social media marketing has no significant influence on the intention to use metaverse technology (H5). This finding underscores the need for further exploration into how promotional efforts on social media platforms impact individuals' willingness to adopt metaverse technology among Thai users. Drawing from research by Puncreobutr et al. [109], it is evident that Thai universities exhibit a high readiness for organizing learning activities in the metaverse era. Factors contributing to this readiness include human development supporting digital competitiveness, internet availability supporting the metaverse, transaction availability on the metaverse, and opportunities to access metaverse technologies.

In this study, we found that consumer engagement significantly influences the intention to use metaverse technology among Thai users (H6). As supported by the accepted hypothesis, heightened engagement with the metaverse concept correlates with a more favorable attitude towards adopting and utilizing this emerging technology. Buhalis et al. [110] emphasize the pivotal role of customer engagement in leveraging metaverse technology for enhanced customer experience and value co-creation. Mishra et al. [111] discuss the impact of technology interfaces on consumer responses, indicating that interactive interfaces foster greater engagement among users. Furthermore, Hilken et al. [112] stress the significance of engagement as a mechanism for elucidating technology-enabled consumer experiences, highlighting its role in driving technology adoption.

## Serial mediation analysis

In our study, social media marketing (SMM) emerges as a significant predictor of the Intention to Use metaverse technology (IU), operating through the intermediary constructs of UTAUT2 and Consumer Engagement (CE) (**Fig 3**). This indicates that while SMM has a notable influence on IU, its impact is fully mediated through UTAUT2 and CE. Essentially, the effect of SMM on IU operates indirectly through these constructs, underscoring the importance of considering the complex interplay between variables in understanding user behaviors and outcomes. This finding is consistent with previous research, such as that conducted by Shien et al. [113], who examined the impact of SMM on young consumers' purchase intentions in Malaysia and highlighted the mediating role of consumer engagement.

Similarly, Emini & Zeqiri [114] explored the mediating role of brand awareness and brand engagement in the relationship between SMM and purchase intention. Their findings revealed

a positive indirect impact of SMM on purchase intention, with brand awareness and brand engagement acting as full mediators. Additionally, Bommer et al. [115] underscored the importance of consumer behavior research in understanding the drivers of consumer actions. By integrating constructs related to cost and consumer behavior into the UTAUT2 model, researchers can gain insights into how consumer engagement and perceived value influence technology adoption, emphasizing the role of consumer engagement in shaping intentions to use and adopt new technologies.

Moreover, Herrero et al. [116] revised the UTAUT2 model to explore the adoption of social networking sites for sharing user-generated content. Their study emphasized the significance of factors such as performance expectancy, hedonic motivation, and habit in influencing the behavioral intention to use technology, which can be influenced by SMM strategies. Analyzing these studies provides insights into how UTAUT2 and consumer engagement act as mediators between SMM and the intention to use technology. Understanding the interaction between SMM strategies and UTAUT2, as well as consumer engagement constructs, can offer valuable insights into how businesses can effectively utilize social media to drive technology adoption intentions.

## Moderated mediation process

This study identified age and experience as critical moderators in the relationship between social media marketing and the intention to use metaverse technology among Thai users, as demonstrated in **Table 7**. Our findings offer valuable insights for both practitioners and researchers, underscoring the necessity of tailoring interventions and strategies to accommodate the diverse characteristics and experiences of users. Previous studies have underscored the significance of age and experience as moderators in this relationship. For example, Venkatesh et al. [117] introduced the Unified Theory of Acceptance and Use of Technology (UTAUT), which identifies age as one of the moderators influencing the prediction of behavioral intention to use technology. Venkatesh & Davis [118] discovered that age, gender, and experience play a moderating role in the relationship between situational constraints (such as performance expectancy, effort expectancy, social influence, and facilitating conditions) and the intention to use technology. Similarly, several pieces of literature have emphasized the importance of age and gender as moderators in studies related to behavioral intentions and technology usage within the context of social media marketing [119–122].

## Theoretical and practical implications

Theoretical implications of the study are substantial, particularly in refining the unified theory of acceptance and use of technology 2 (UTAUT2) model for the metaverse context. By introducing a higher-order reflective-formative construct of UTAUT2, combining seven lower-order constructs, the research endeavors to streamline the model, mitigate postulated relationships, and tackle collinearity issues. This methodological refinement contributes significantly to the advancement of theoretical frameworks in technology acceptance research, offering a more subtle understanding of user behavior within virtual environments. Moreover, the study delves into the dimensions of metaverse engagement, encompassing performance expectancy, effort expectancy, social influence, facilitating conditions, hedonic motivation, habit, and price value. This comprehensive analysis provides insights into the intricate factors influencing users' intention to utilize metaverse platforms, enriching existing literature on virtual environments and user adoption behavior. Additionally, the research showcases methodological advances in measurement models, demonstrating meticulous evaluation techniques for assessing convergent validity, discriminant validity, and reliability. These methodological insights

offer valuable contributions to the field, guiding researchers in handling reflective-formative constructs and higher-order constructs in structural equation modeling with precision and rigor.

Practical implications of the study extend to various facets of research design, data collection, and analytical techniques, offering actionable insights for practitioners in the technology industry. Firstly, the study provides guidance on questionnaire design and recruitment strategies tailored to the metaverse context, emphasizing the importance of pretesting and reverse encoding to rectify response bias. Leveraging social media platforms and chat features for participant recruitment showcases effective strategies for reaching potential respondents and enhancing survey engagement. Secondly, the research underscores the significance of targeted sampling strategies and sample size considerations in data collection, particularly in geographic regions with significant market potential. By employing power analysis techniques, researchers can ensure robust statistical analysis and adequate representation of the target population. Furthermore, practitioners can derive valuable insights from the detailed analysis of structural models, mediation processes, and moderated mediation effects, aiding strategic decision-making in designing marketing campaigns and user engagement initiatives within the metaverse. Finally, the integration of Partial Least Squares Structural Equation Modeling (PLS-SEM) with artificial neural networks (ANN) presents a comprehensive approach to data analysis, capturing both linear and non-linear relationships among variables. This integration enhances the accuracy and predictive power of models, offering actionable insights for stakeholders in the technology industry to optimize user experiences and drive innovation in virtual environments.

## Conclusions

The study explored the adoption intentions of Thai users towards metaverse technology within social media marketing (SMM) contexts using the unified theory of acceptance and use of technology (UTAUT2) framework. This study proposed a reflective-formative construct combining seven lower-order UTAUT2 constructs and validates it through questionnaire design and data collection from internet users in Bangkok and surrounding provinces. The analysis, conducted via Partial Least Squares Structural Equation Modeling (PLS-SEM), confirms the reliability and validity of the research model. Overall, the findings suggest that the models used for structural model analysis, serial mediation analysis, and moderated mediation process have good predictive potential and supports its robustness in forecasting Thai users' intention to use metaverse technology. The analyses revealed significant relationships among constructs, with SMM exerting the strongest influence on UTAUT2, followed by its effects on consumer engagement (CE) and intention to use metaverse technology (IU). Moreover, moderation effects of age and technology experience on the SMM-UTAUT2 relationship are identified. Integration of PLS-SEM and artificial neural network (ANN) models showcases the accuracy of the proposed framework, highlighting UTAUT2 as the primary predictor of IU in Thai users' adoption of metaverse technology for SMM. However, limitations exist, such as the focus on a specific demographic and the lack of exploration into Thailand's cultural and economic landscape. Future research could delve into additional moderators or mediators influencing the SMM-UTAUT2 relationship and incorporate longitudinal or qualitative methods to deepen understanding. These avenues could enhance our comprehension of metaverse technology adoption in marketing contexts.

## Supporting information

**S1 File. Questionnaire.**
(DOCX)

**S1 Table. Demographic characteristics of the respondents.**
(DOCX)

**S2 Table. Higher order construct (UTAUT2).**
(DOCX)

**S3 Table. ANN- RMSE results.**
(DOCX)

**S4 Table. Sensitivity analysis.**
(DOCX)

## Author Contributions

**Conceptualization:** Chaloempon Sritong, Teerapong Teangsompong.

**Data curation:** Chaloempon Sritong, Teerapong Teangsompong.

**Formal analysis:** Weerachon Sawangproh, Teerapong Teangsompong.

**Investigation:** Weerachon Sawangproh, Teerapong Teangsompong.

**Methodology:** Chaloempon Sritong, Weerachon Sawangproh, Teerapong Teangsompong.

**Project administration:** Teerapong Teangsompong.

**Resources:** Chaloempon Sritong, Teerapong Teangsompong.

**Software:** Teerapong Teangsompong.

**Validation:** Chaloempon Sritong, Weerachon Sawangproh, Teerapong Teangsompong.

**Visualization:** Teerapong Teangsompong.

**Writing – original draft:** Chaloempon Sritong, Weerachon Sawangproh, Teerapong Teangsompong.

**Writing – review & editing:** Chaloempon Sritong, Weerachon Sawangproh, Teerapong Teangsompong.

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
