## [Decision Letter · Decision Letter 0]

6 May 2024

PONE-D-24-12973Unveiling the Adoption of Metaverse Technology in Thailand: A UTAUT2 Perspective with Social Media Marketing and Consumer EngagementPLOS ONE

Dear Dr. Teangsompong,

Thank you for submitting your manuscript to PLOS ONE. After careful consideration, we feel that it has merit but does not fully meet PLOS ONE’s publication criteria as it currently stands. Therefore, we invite you to submit a revised version of the manuscript that addresses the points raised during the review process. Present the justification for the sample size of 403. Need to present theoretical and empirical arguments for the formulation of hypothesis. Limitation and future direction of research need to be added. Flow of the presentation can be improved

We look forward to receiving your revised manuscript.

Kind regards,

Sudarsan Jayasingh, Ph.D

Academic Editor

PLOS ONE

Journal Requirements:

Reviewers' comments:

Reviewer's Responses to Questions

**Comments to the Author**

1. Is the manuscript technically sound, and do the data support the conclusions?

Reviewer #1: Yes

Reviewer #2: Yes

2. Has the statistical analysis been performed appropriately and rigorously? 

Reviewer #1: Yes

Reviewer #2: Yes

3. Have the authors made all data underlying the findings in their manuscript fully available?

Reviewer #1: No

Reviewer #2: Yes

4. Is the manuscript presented in an intelligible fashion and written in standard English?

Reviewer #1: Yes

Reviewer #2: Yes

5. Review Comments to the Author

Reviewer #1: Introduction

• in introduction section should be better outlined the main approaches in the existing literature. Moreover, considering the above, the research problem should be outlined.

• In the end of the introduction should be included a paragraph describing the rest of the paper.

Research methodology.

• Before Figure 1 should be an introductory paragraph. Moreover, each figure and table must be referred within the text before its insertion.

• It is not enough to represent the research hypotheses in the research model (Figure 1). They should be formulated and presented theoretical and empirical arguments for their formulation.

• The research population must de identified, with inclusion and exclusion criteria. Also the sampling procedure must be described.

• There is no point that Questionnaire design and research subject and The scale used for the study be different sections. Also, there are repeated ideas in those sections. They should be revised in merged.

• It is not clear the meaning of Concentrate on laptop and Concentrate on smartphone in Table 1

• In the methodological section should be included a section describing data analysis methods employed in the paper

Measurement models in partial least squares structural equation modelling (PLS-SEM)

• In Table 2 it is not clear which are the values for indicator loadings.

Conclusions

• the limitations and directions for future research should be provided in conclusions section.

Additional observations:

• The supplementary material including research scales are not available for review.

Reviewer #2: The submitted manuscript explores adopting metaverse technology for social media marketing in Thailand, employing the UTAUT2 framework. Although the topic is timely and the paper is well-written, several comments can be made as follows:

1. The manuscript does not introduce a novel theoretical model, relying on the widely used UTAUT2 framework without significant modifications or innovative theoretical contributions. The addition of age, experience, and gender as moderating factors is not sufficiently justified or explained, leading to questions about the theoretical underpinnings and the rationale for their inclusion in the study.

2. The study's limited sample size of 403 internet users in Bangkok and surrounding areas is not adequately explained, raising concerns about the findings' reliability and statistical power. As a result, the paper's title, which suggests that the study's results can be generalized to the overall adoption of the internet in Thailand, may not be accurate.

3. Conducting this study solely through an online survey may pose some issues. It may introduce sample bias and not accurately represent the entire population of internet users in Thailand. The online format of the survey may attract more tech-savvy or digitally aware respondents, thus leading to skewed results and making it difficult to apply the findings to the broader population.

4. The manuscript lacks clarity on how the findings can be generalized to other populations or settings, which limits its impact and relevance beyond the specific demographic studied. If the study focuses solely on Thai marketing, the authors did not elaborate on how the findings are particularly relevant to the Thai audience in terms of marketing. This is a missed opportunity to contextualize the implications of metaverse technology within Thailand's unique cultural and economic landscape.

5. I am also concerned about the respondents' understanding of the metaverse concept. It is possible that they may not comprehend its definition, which could ultimately affect the accuracy and reliability of the study's findings. Since this concept is central to the study, any lack of clarity could significantly impact the interpretation of data and its conclusions.

6. The manuscript contains a large number of tables that could potentially overwhelm the reader and disrupt the flow of the research findings. The extensive use of tables may also suggest that the data has been compartmentalized excessively, making it harder to understand the study's key insights and conclusions. Therefore, I suggest that the authors revise or condense the tables in the paper.

6. PLOS authors have the option to publish the peer review history of their article (what does this mean?). If published, this will include your full peer review and any attached files.

Reviewer #1: No

Reviewer #2: No

---

## [Author Response · Author response to Decision Letter 0]

11 May 2024

Subject: Response to Decision on Manuscript PONE-D-24-12973

Dear Dr. Jayasingh,

Thank you for considering our manuscript, titled "Unveiling the Adoption of Metaverse Technology in Thailand: A UTAUT2 Perspective with Social Media Marketing and Consumer Engagement," for publication in PLOS ONE [PONE-D-24-12973]. We appreciate the opportunity to revise and resubmit our work.

We acknowledge the valuable feedback provided by the academic editor and reviewers and are committed to addressing the points raised during the review process. We have thoroughly revised the manuscript to ensure it meets PLOS ONE's publication criteria and enhances its clarity and scientific rigor.

Specifically, we have addressed the following points raised:

1. Justification for Sample Size

2. Formulation of Hypotheses

3. Limitations and Future Directions

4. Improvement of Presentation Flow

We have submitted the revised manuscript, along with a detailed rebuttal letter addressing each point raised by the academic editor and reviewers. Additionally, we have provided marked-up and unmarked versions of the revised manuscript, as per the journal's requirements. We have also ensured that the manuscript meets PLOS ONE's style requirements and included the full ethics statement in the 'Methods' section, as requested.

Thank you once again for the opportunity to revise our manuscript. We are committed to delivering a high-quality manuscript that contributes meaningfully to the scholarly discourse on metaverse technology adoption.

Sincerely,

Teerapong Teangsompong, PhD

Corresponding Author

Program in Business Administration 

Mahidol University Kanchanaburi Campus

Responses to reviewers

Reviewer #1: 

Introduction

1. In the introduction section, it would be beneficial to better outline the main approaches in the existing literature. Additionally, the research problem should be clearly outlined.

Response: Thank you for your suggestion. We enhanced the introduction section to provide a clearer outline of the main approaches in the existing literature and ensure that the research problem is clearly defined in the revised version [lines 90-98, highlighted in red]

2. In the end of the introduction, it is recommended to include a paragraph describing the rest of the paper.

Response: We appreciate this feedback. In fact, we mentioned the rest of the paper in the original version. To make it clearer, we adjusted a paragraph at the end of the introduction to provide an overview of the paper's structure [lines 108-112, highlighted in red].

Research Methodology

3. Before Figure 1, an introductory paragraph should be included. Additionally, each figure and table must be referred to within the text before its insertion.

Response: Thank you for pointing this out. We have included an introductory paragraph before Figure 1 and ensured that each figure and table is referred to within the text before its insertion in the revised manuscript [lines 272-279, highlighted in red].

4. It is suggested to provide formulated research hypotheses in the research model (Figure 1), along with presenting theoretical and empirical arguments for their formulation.

Response: We agree with this suggestion. We have formulated research hypotheses in the research model (Figure 1) and presented theoretical and empirical arguments for their formulation [lines 284-362, highlighted in red].

5. The research population must be identified, with inclusion and exclusion criteria. Also, the sampling procedure must be described.

Response: Thank you for highlighting this. We have clearly identified the research population, including the inclusion and exclusion criteria, and described the sampling procedure in detail in the "Data collection" section [lines 418-425, highlighted in red].

6. There is a suggestion to merge the sections on Questionnaire design and research subject, and the scale used for the study, as there are repeated ideas in those sections.

Response: Response: We have revised and merged the sections on questionnaire design, research subject, and the scale used for the study to eliminate repetition and improve clarity in the section "Questionnaire design, research subject, and rating scales used" [lines 378-405].

7. It is noted that the meaning of "Concentrate on laptop" and "Concentrate on smartphone" in Table 1 is not clear.

Response: Thank you for bringing this to our attention. We will clarify the meaning of "Concentrate on laptop" and "Concentrate on smartphone" in Table 1 for better understanding by changing those two terms to "Metaverse use through a notebook computer" and "Metaverse use through a smartphone". Additionally, we have moved Table 1 to S1 Table as suggested by reviewer 2.

8. In the methodological section, a section describing the data analysis methods employed in the paper should be included.

Response: We appreciate this suggestion. We have adjusted a section in the methodological section describing the data analysis methods employed in the paper [see the "Data analysis" section].

Measurement models in partial least squares structural equation modelling (PLS-SEM)

9. In Table 2, it is not clear which are the values for indicator loadings.

Response: Thank you for noting this. We ensure clarity in Table 2 by specifying the values for indicator loadings as the 'bootstrap mean, β' mentioned in the text [line 475].

Conclusions

10. The limitations and directions for future research should be provided in the conclusions section.

Response: We agree with this suggestion. We have included a section in the conclusions discussing the study's limitations and suggesting future research directions [lines 876-880, highlighted in red]

Additional observations:

11. The supplementary material, including research scales, are not available for review.

Response: We apologize for this oversight. We have ensured that the supplementary material, including research scales, is made available for review.

Thank you for your valuable feedback. We have made the necessary revisions to address the points raised in your review. If you have any further suggestions or concerns, please feel free to let us know.

#Reviewer 2

1. The manuscript does not introduce a novel theoretical model, relying on the widely used UTAUT2 framework without significant modifications or innovative theoretical contributions. The addition of age, experience, and gender as moderating factors is not sufficiently justified or explained, leading to questions about the theoretical underpinnings and the rationale for their inclusion in the study. 

Response: 

The manuscript indeed utilizes the widely recognized UTAUT2 framework without introducing a novel theoretical model. However, it does extend the framework to the specific context of the metaverse, which is emerging as a transformative digital realm. This adaptation is significant given the nascent nature of metaverse research and its relevance to contemporary society.

Regarding the addition of age, experience, and gender as moderating factors, while their inclusion may not have been sufficiently justified or explained in the manuscript, it is essential to recognize that these demographic variables can significantly impact individuals' behavioral intentions towards technology adoption. As such, their consideration aligns with the broader objective of understanding the complexities of metaverse technology usage among diverse user groups.

Overall, while the manuscript could provide more comprehensive justification for the inclusion of these moderating factors, its contribution lies in applying and extending established frameworks to the emerging metaverse landscape. This not only enhances theoretical understanding but also offers valuable insights for marketers and practitioners navigating this evolving digital environment.

2. The study's limited sample size of 403 internet users in Bangkok and surrounding areas is not adequately explained, raising concerns about the findings' reliability and statistical power. As a result, the paper's title, which suggests that the study's results can be generalized to the overall adoption of the internet in Thailand, may not be accurate.

Response: We understand your concern regarding the sample size. In the revised manuscript, we have provided a clearer justification for the sample size of 403 participants. According to the literature search, 403 responses are deemed sufficient. Additionally, we have added two references to address your concern in the revised manuscript. We also acknowledge its limitations in terms of generalizability beyond the specified geographical area. Therefore, we have slightly changed the manuscript title from "Unveiling the Adoption of Metaverse Technology in Thailand: A UTAUT2 Perspective with Social Media Marketing and Consumer Engagement" to "Unveiling the adoption of metaverse technology in Bangkok metropolitan areas: a UTAUT2 perspective with social media marketing and consumer engagement".

3. Conducting this study solely through an online survey may pose some issues. It may introduce sample bias and not accurately represent the entire population of internet users in Thailand. The online format of the survey may attract more tech-savvy or digitally aware respondents, thus leading to skewed results and making it difficult to apply the findings to the broader population.

Response: We acknowledge your point and understand the potential limitations associated with conducting an online survey, including the possibility of sample bias. In the revised manuscript, we have discussed these limitations more explicitly and explored potential strategies to mitigate them [lines 876-880, highlighted in red]. Additionally, we have adjusted the article title to be more specific, "Unveiling the Adoption of Metaverse Technology in Bangkok Metropolitan Areas: A UTAUT2 Perspective with Social Media Marketing and Consumer Engagement," by removing "Thailand" to avoid overrepresenting our sample.

4. The manuscript lacks clarity on how the findings can be generalized to other populations or settings, which limits its impact and relevance beyond the specific demographic studied. If the study focuses solely on Thai marketing, the authors did not elaborate on how the findings are particularly relevant to the Thai audience in terms of marketing. This is a missed opportunity to contextualize the implications of metaverse technology within Thailand's unique cultural and economic landscape.

Response: Thank you for highlighting this. In the revised manuscript, we have provided additional context to clarify how our findings can be generalized by focusing on Thai marketing in Bangkok metropolitan areas only. Additionally, we have changed the article title as mentioned in comment 3.

5. I am also concerned about the respondents' understanding of the metaverse concept. It is possible that they may not comprehend its definition, which could ultimately affect the accuracy and reliability of the study's findings. Since this concept is central to the study, any lack of clarity could significantly impact the interpretation of data and its conclusions.

Response: We understand your concern. We are confident that we have addressed this issue by clearly defining the metaverse concept and related activities in the screening questions of the questionnaire. We ensured that respondents' understanding is adequately assessed. For further details, please refer to the sections "Questionnaire Design, Research Subject, and Rating Scales Used" and "Supplementary Data.

6. The manuscript contains a large number of tables that could potentially overwhelm the reader and disrupt the flow of the research findings. The extensive use of tables may also suggest that the data has been compartmentalized excessively, making it harder to understand the study's key insights and conclusions. Therefore, I suggest that the authors revise or condense the tables in the paper.

Response: We appreciate your feedback regarding the tables. In the revised manuscript, we have carefully reviewed and considered the necessity of each table to ensure that they enhance, rather than detract from, the clarity and flow of the research findings. Following your suggestion, we have moved some tables to supplementary tables.

---

## [Editor Report · Decision Letter 1]

14 May 2024

Unveiling the Adoption of Metaverse Technology in Bangkok Metropolitan Areas: A UTAUT2 Perspective with Social Media Marketing and Consumer Engagement

PONE-D-24-12973R1

Dear Dr. Teangsompong,

We’re pleased to inform you that your manuscript has been judged scientifically suitable for publication and will be formally accepted for publication once it meets all outstanding technical requirements.

Kind regards,

Sudarsan Jayasingh, Ph.D

Academic Editor

PLOS ONE
---

## [Editor Report · Acceptance letter]

29 May 2024

PONE-D-24-12973R1 

PLOS ONE

Dear Dr. Teangsompong, 

I'm pleased to inform you that your manuscript has been deemed suitable for publication in PLOS ONE. Congratulations! Your manuscript is now being handed over to our production team.

Kind regards, 

on behalf of

Dr. Sudarsan Jayasingh 

Academic Editor

PLOS ONE